# Deep learning model DeepNeo predicts neointimal tissue characterization using optical coherence tomography
Valentin Koch [1,2,3], Olle Holmberg[1,2,4], Edna Blum[5], Ece Sancar[1,2], Alp Aytekin[5], Masaru Seguchi[5], Erion Xhepa[5], Jens Wiebe[5], Salvatore Cassese[5], Sebastian Kufner[5], Thorsten Kessler[5,6], Hendrik Sager [5,6], Felix Voll [5], Tobias Rheude[5], Tobias Lenz[5], Adnan Kastrati[5,6], Heribert Schunkert [5,6], Julia A. Schnabel [1,2,7], Michael Joner[5,6] ✉, Carsten Marr[1] ✉ & Philipp Nicol[5]

## Abstract

**Background** Accurate interpretation of optical coherence tomography (OCT) pullbacks is critical for assessing vascular healing after percutaneous coronary intervention (PCI). Manual analysis is time-consuming and subjective, highlighting the need for a fully automated solution.

**Methods** In this study, 1148 frames from 92 OCT pullbacks were manually annotated to classify neointima as homogeneous, heterogeneous, neoatherosclerosis, or not analyzable on a quadrant level. Stent and lumen contours were annotated in 305 frames for segmentation of the lumen, stent struts, and neointima. We used these annotations to train a deep learning algorithm called DeepNeo. Performance was further evaluated in an animal model (male New Zealand White Rabbits) of neoatherosclerosis using co-registered histopathology images as the gold standard.

**Results** DeepNeo demonstrates a strong classification performance for neointimal tissue, achieving an overall accuracy of 75%, which is comparable to manual classification accuracy by two clinical experts (75% and 71%). In the animal model of neoatherosclerosis, DeepNeo achieves an accuracy of 87% when compared with histopathological findings. For segmentation tasks in human pullbacks, the algorithm shows strong performance with mean Dice overlap scores of 0.99 for the lumen, 0.66 for stent struts, and 0.86 for neointima.

**Conclusions** To the best of our knowledge, DeepNeo is the first deep learning algorithm enabling fully automated segmentation and classification of neointimal tissue with performance comparable to human experts. It could standardize vascular healing assessments after PCI, support therapeutic decisions, and improve risk detection for cardiac events.

## Plain language summary

Optical coherence tomography (OCT) is an imaging test used to detect blood vessel healing after a minimally invasive procedure that widens blocked heart arteries. However, this analysis is performed manually, which is time-consuming and can lead to inconsistencies in diagnosis between clinicians. In this study, we developed a computer-aided tool called DeepNeo that can analyze OCT images of blood vessels and detect vascular healing automatically. We trained the computer-aided technology using manually annotated OCT images and tested its performance on both human and animal data. DeepNeo performed to a similar degree as clinicians. Our findings suggest that DeepNeo can standardize and automate OCT image analysis, potentially improving the efficiency of vascular healing assessments and aiding clinical decision-making to reduce the risk of future cardiac events.

Interventional revascularization by percutaneous coronary intervention (PCI) with stent implantation is an important treatment option for patients with obstructive coronary artery disease[1]. Despite advancements in the field of PCI, including refinement of contemporary drug-eluting

stent (DES) technology, a proportion of patients still experience stent-related events such as in-stent restenosis or stent thrombosis in the long-term[2]. The development of mature and healthy stent-covering neointima is critical to prevent these adverse events. However, delayed vascular

[1]Institute of AI for Health, Helmholtz Munich—German Research Center for Environmental Health, Munich, Germany. [2]School of Computation and Information Technology, Technical University of Munich, Munich, Germany. [3]Munich School for Data Science, Munich, Germany. [4]Helsing GmbH, Munich, Germany. [5]German Heart Centre Munich, Technical University of Munich, Munich, Germany. [6]German Center for Cardiovascular Research, Partner Site Munich Heart Alliance, Munich, Germany. [7]School of Biomedical Engineering and Imaging Sciences, King's College London, London, UK. ✉e-mail: joner@dhm.mhn.de; carsten.marr@helmholtz-munich.de

healing can impair neointimal development and contribute to stent failure[3,4]. Hence, immature or diseased neointima play an important role in a substantial portion of cases with stent failure[2,5]. Optical coherence tomography (OCT), as a high-resolution intravascular imaging modality, provides detailed visualization of the coronary vasculature and can be used to assess the mode of stent failure[1,6,7]. Using OCT, neointima can be visualized in vivo and characterized as either homogenous or heterogenous. Previous studies have shown that homogenous neointimal tissue has a favorable phenotype, while heterogenous neointimal tissue may be associated with de novo atherosclerosis (neoatherosclerosis) and a worse clinical outcome[8–13]. Therefore, accurate detection and distinction of neointimal tissue is an important step in identifying patients at risk for stent failure. However, manual evaluation of OCT images is time-consuming and highly dependent on clinician experience, which can limit clinical availability and transferability[13]. Moreover, the visual interpretation of OCT images by clinicians in their daily practice may result in missing or underestimating relevant pathological changes. Hence, more standardized approaches to OCT image analysis are necessary. Deep learning has the potential to greatly assist clinicians in accurately diagnosing patients through the analysis of medical images[14,15]. In intravascular OCT imaging, deep learning has been successfully used to characterize native atherosclerotic lesions[16,17]. In this study, we present the fully automated deep learning-based algorithm DeepNeo, that enables quick and accurate automated segmentation and classification of neointimal tissue characteristics. DeepNeo demonstrates strong classification performance, achieving an overall accuracy of 75% for neointimal tissue, comparable to expert manual classification. In an animal model of neoatherosclerosis, it reaches 87% accuracy against histopathological findings. Additionally, DeepNeo exhibits robust segmentation performance in human OCT pullbacks, with mean Dice overlap scores of 0.99 for the lumen, 0.66 for stent struts, and 0.86 for neointima.

## Methods

This study has been approved by the ethical board of the Technical University of Munich, Germany in accordance with local regulations (No. 2023-143-S-NP). Informed consent of the patients undergoing coronary angiography and OCT was obtained during the clinical routine. The rabbit model was approved by the government ("Regierung von Oberbayern", file number 55.2.1.54-2532-40-16) and was in accordance with the German Animal Welfare Act (version May 18, 2006, amended on March 29, 2017) as well as directive 2010/63/EU of the European Parliament on the protection of animals used for scientific purposes. A patent application describing the technology has been filed with the European Patent Office (Application 23 179 433).

### Data acquisition

1148 OCT images from 92 patients who underwent clinically indicated coronary angiography and in-stent intravascular imaging with OCT at the German Heart Center Munich were collected. OCT imaging was performed according to current guidelines[18] using a commercially available OCT system (Abbott Vascular, Santa Clara, CA). The baseline characteristics of patients are provided in Table 1.

### Segmentation of neointima, lumen, and stent struts

Lumen contour and stent struts were manually annotated in 305 OCT frames from a subset (40 of the 92 pullbacks) using the freeware tool LabelMe (available at http://labelme.csail.mit.edu/Release3.0/) to enable automated segmentation of stent struts, lumen, and neointimal area (see Fig. 1a). Segmentation allows analysis of patient characteristics such as average neointima thickness, detection of areas of uncovered stent struts, or the localization of the minimal lumen diameter in the stent. Also, segmentation masks allow the calculation of the center of the lumen, which is used to cut frames into quadrants. To assess the performance of DeepNeo for the segmentation of neointima, lumen, and stent struts, we employed a 5-fold cross-validation approach. This involved randomly dividing the dataset of 305 OCT frames into five equal folds (parts), with one fold used as a test set and the remaining four folds split into three training sets and one validation set. We repeated this process five times, with each fold used as the test set once. To prevent information leaks, frames from any unique patient were assigned to the same fold. The validation set was used to adjust hyperparameters that determine the model architecture and training procedure and choose the most suitable model.

### Neointima classification

Manual quadrant annotation of OCT frames every 1 mm (every fifth frame) was performed for all pullbacks or in adjacent suitable frames when image quality was insufficient. Neointimal tissue was classified using a quadrant-based nominal character scoring system as previously described[19]: clockwise and starting at 12 o'clock, every frame was divided into four quadrants (see Fig. 1a), with the center of the lumen as the dividing point. Each quadrant was then independently classified according to its predominant neointimal appearance into one of four classes: homogenous neointima (uniform light reflection without localized areas of stronger or weaker backscattering properties), heterogenous neointima (consisting of a focal variation of the backscattering pattern, including patterns described as layered), neoatherosclerosis (containing neointimal foam cells, fibroatheroma or calcifications)[20,21], or not analyzable (quadrants with uncovered struts or side-branch openings). In quadrants with more than one tissue type, the most severe neointimal tissue type was scored. Examples of neointimal tissue types are illustrated in Fig. 1b. Expert A manually classified a total of 1148 frames (i.e., 4592 single quadrants) from 92 pullbacks. From the total of 1148 OCT frames derived from 92 pullbacks, we allocated 936 frames (originating from 66 pullbacks) to the training set. The validation set comprised 108 frames from 9 pullbacks, while the test set included 104 frames from 17 pullbacks. This test set was specifically used to assess inter-observer variability and the final performance of DeepNeo, with frames being independently analyzed by experts B and C. The split was made by the patient, e.g. any patient's frames are only contained in one of the train/validation/test splits. The Fleiss Kappa score for the three independent experts was 0.654 for the test set.

### Animal model for neointima classification

As previously published, male New Zealand White rabbits were obtained at an age of 3–4 months from Charles River, France and underwent stent implantation in iliac arteries and repeated balloon denudation under a hypercholesterolemic diet, promoting early neoatherosclerotic lesion formation over 161 days[22]. Animals were under a 12-h light/dark cycle and were fed a 1% cholesterol diet ad libitum (Altromin Spezialfutter GmbH, Lage, Germany) for 7 days prior to balloon denudation of the iliac arteries, followed by stent implantation. After another 4 weeks of a high-cholesterol diet (1%), animals were switched to a 0.025% cholesterol diet (Altromin Spezialfutter GmbH) at day 35 and were continued on this diet until euthanasia. OCT imaging and histopathological analysis of stented segments were performed using co-registration of both modalities, where OCT frames were aligned with matching histopathology frames. The co-registration process was based on the lumen contour and the position of the stent struts in the corresponding section, as previously described[22]. Histopathology frames were divided into quadrants and scored according to the predominant tissue characteristic in each quadrant. To ensure consistency and comparability across the scoring process, we utilized a nominal character scoring system similar to that employed by DeepNeo. Specifically, a "homogeneous" score was assigned to frames demonstrating healthy neointima with a predominance of smooth muscle cells, whereas frames demonstrating infiltration with foam cells were assigned a "neoatherosclerosis" score. Frames showing deposition of fibrin, hypocellular neointima, or peristrut hemorrhage were assigned a "heterogeneous" score. It should be noted that the rabbit dataset was entirely distinct from the human dataset. DeepNeo analyzed OCT pullbacks from 12 rabbits (15 frames), and its neointimal tissue predictions were compared to the co-registered histopathology findings.

**Table 1 | Baseline data of OCT data set**

| N | | 92/92 (100.0%) |
|---|---|---|
| Age in years | | 67.3 (±9.3) |
| Gender | Female | 13/92 (16.3%) |
| | Male | 77/92 (83.7%) |
| Cardiovascular risk factor | Smoker | 17/92 (18.5%) |
| | Hypercholesterolemia | 65/92 (70.7%) |
| | Hypertension | 88/92 (95.7%) |
| | Diabetes mellitus | 40/92 (43.5%) |
| Left ventricular function (ejection fraction) | Normal | 53/92 (57.6%) |
| | Mildly reduced | 20/92 (21.7%) |
| | Reduced | 18/92 (19.6%) |
| | Severely reduced | 1/92 (1.1%) |
| Coronary artery disease | One-vessel | 12/92 (13.0%) |
| | Two-vessel | 17/92 (18.5%) |
| | Three-vessel | 63/92 (68.5%) |
| Clinical presentation | Stable angina | 53/92 (57.6%) |
| | Silent ischemia | 21/92 (22.8%) |
| | Non-ST-Elevation myocardial infarction | 7/92 (7.6%) |
| | Unstable angina | 10/92 (10.9%) |
| | ST-Elevation myocardial infarction | 1/92 (1.1%) |
| Target vessel | Left anterior descending artery | 42/92 (45.7%) |
| | Left coronary artery | 2/92 (2.2%) |
| | Left circumflex artery | 26/92 (31.5%) |
| | Right coronary artery | 22/92 (23.9%) |
| Restenosis morphology | Complete occlusion | 3/92 (3.3%) |
| | Diffuse beyond stent | 1/92 (1.1%) |
| | Diffuse intrastent | 25/92 (27.2%) |
| | Focal body | 43/92 (46.7%) |
| | Focal margin | 6/92 (6.5%) |
| | Multifocal | 6/92 (6.5%) |
| | No restenosis | 8/92 (8.7%) |
| Index stent interval in days | | 1356.6 (±1477.4) |
| Index stent type | Biodegradable polymer-eluting stent | 5/92 (5.4%) |
| | Bare-metal stent | 4/92 (4.3%) |
| | Biodegradable polymer sirolimus-eluting stent | 8/92 (8.7%) |
| | Drug-eluting stent | 4/92 (4.3%) |
| | Everolimus-eluting stent | 46/92 (50.0%) |
| | Polymer-free sirolimus-eluting stent | 3/92 (3.3%) |
| | Sirolimus-eluting stent | 3/92 (3.3%) |
| | Zotarolimus-eluting stent | 3/92 (3.3%) |
| | Unknown | 16/92 (17.4%) |
| Lesion length in mm | | 11.5 (±6.2) |

Patient baseline data (n = number of patients with characteristics, N = total number of patients) reported as n/N (%) or mean (±standard deviation).

## Algorithm architecture

We employed two deep neural networks, trained separately and combined during inference, to (i) segment lumen, stent struts, and neointima and (ii) classify the neointima in each quadrant of an OCT frame (see Fig. 1c). To segment stent struts, neointima, and lumen, a UNet++ was used[23]. Details about the training of the segmentation network can be found in the supplemental methods in the "Details of segmentation network" section. For the classification of the quadrants, a ResNet-18[24] network was used. To train the classification network, we divided each frame into four quadrants, using the segmentation generated by the UNet++ to determine the center of the lumen, and rescaled them to a resolution of $224 \times 224$ pixels. Model calibration was achieved through temperature sharpening and fusion of the surrounding quadrants' prediction[25]. Details on the need for calibration and the formula used can be found in the supplemental methods under the "Details of classification network" section.

## Statistics and reproducibility

Statistical analyses were performed using metrics appropriate for segmentation and classification tasks. A 5-fold cross-validation approach was employed to evaluate segmentation performance on 305 frames, ensuring that frames from the same patient were confined to a single fold to prevent information leakage. Hyperparameters were optimized on validation sets, and performance was reported on test sets derived from independent patient data.

Sample sizes were defined as 936 frames for training, 108 frames for validation, and 104 frames for testing for classification. Data splits were performed at the patient level to ensure independence between sets. Reproducibility was ensured by setting all random seeds. The rabbit dataset was used as an independent test set to validate the model's generalizability, demonstrating consistent results across species and imaging setups.

## Reporting summary

Further information on research design is available in the Nature Portfolio Reporting Summary linked to this article.

## Results

### Segmentation of neointima, lumen, and stent struts

DeepNeo achieves high accuracy in segmentation of lumen, stent struts, and neointima with a Dice score of 0.99 (±0.02), 0.66 (±0.10), and 0.86 (±0.14), respectively. The frames exhibiting inferior scoring are observed solely in regions characterized by minimal or absent neointima, thereby rendering precise annotation and prediction of the neointimal regions challenging and susceptible to marginal annotation variability (see Fig. 2, low score sample). In Supplemental Table 1, different segmentation networks are compared.

### Neointima classification

We compare the neointimal tissue classification performance of DeepNeo to that of clinical experts by having two additional independent specialists (expert B and expert C) manually label the test set in a blinded fashion. The labels annotated by the most experienced expert A are assumed to be the ground truth and compared to the labels predicted by DeepNeo. DeepNeo achieves an accuracy of 0.75 and a macro F1-Score of 0.74, while expert B has an accuracy of 0.75 with a macro F1-Score of 0.75, and expert C has an accuracy of 0.71 with a macro F1-Score of 0.69, highlighting a high agreement of DeepNeo with the experts, which is similar to the inter-observer agreement. Figure 3 provides a comparison of manual annotations by experts A–C with the automated prediction by DeepNeo. Note that frames with disagreement between experts (Fig. 3a, b, and d) result in lower prediction certainty (thin prediction line) compared to frames with agreement between observers (thick prediction line). A robust correlation is observed between the model's confidence in the predicted class and the probability of a correct prediction (Fig. 4), indicating that DeepNeo is well-calibrated. This is of special importance, as it gives a notion of confidence and thus interpretability that many other algorithms lack. In Supplemental Fig. S1 we show the need for calibration: the uncalibrated version of our model tends to be overly confident, and the correlation between the (uncalibrated) confidence and the true probability is poor. In Supplemental Table 2, different classification models are evaluated.

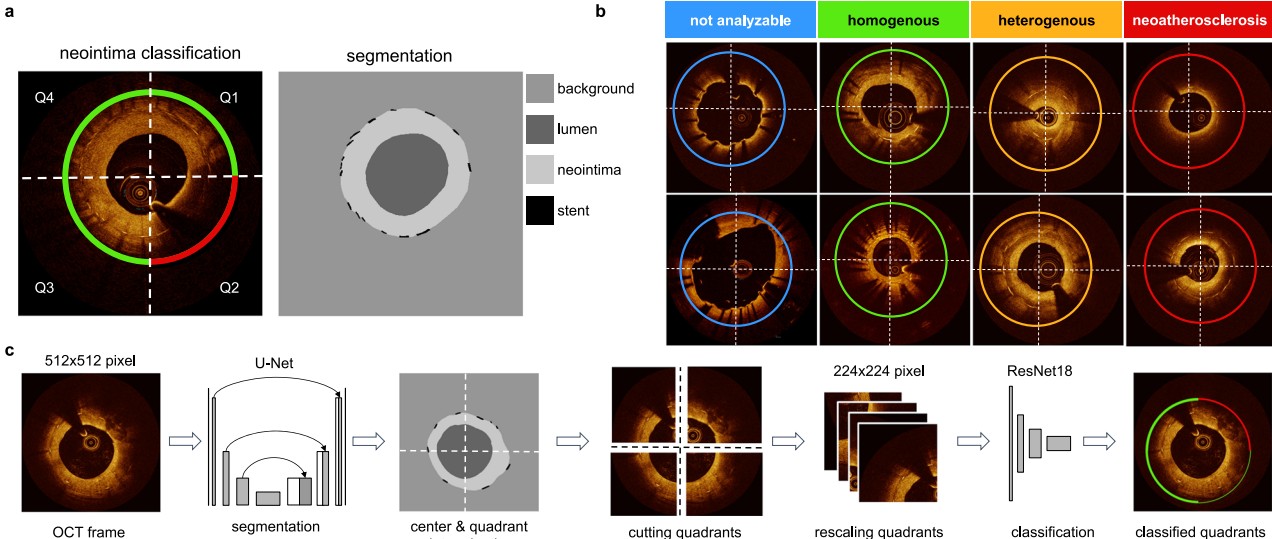

**Fig. 1 | DeepNeo provides neointimal tissue segmentation and classification on the quadrant level. a** OCT frames are divided into four 90° quadrants (Q1–Q4), rotating clockwise from 12 o'clock and individually classified into one of four classes indicated by the circular line color. Vessel lumen, neointima, and stent struts are annotated pixelwise. **b** Representative examples of homogenous, heterogenous, neoatherosclerosis and not analyzable OCT frames used in the study. **c** DeepNeo architecture: A frame is given as input to a U-Net to get a segmentation mask. This allows the calculation of the center of the lumen and the division of the OCT frame into 4 quadrants at the center, which are then each resized to a size of 224 × 224 pixels before going through the classification network (ResNet-18). The colored quarter-circles show the predicted class for each quadrant, line thickness indicates model certainty (thick line = high certainty).

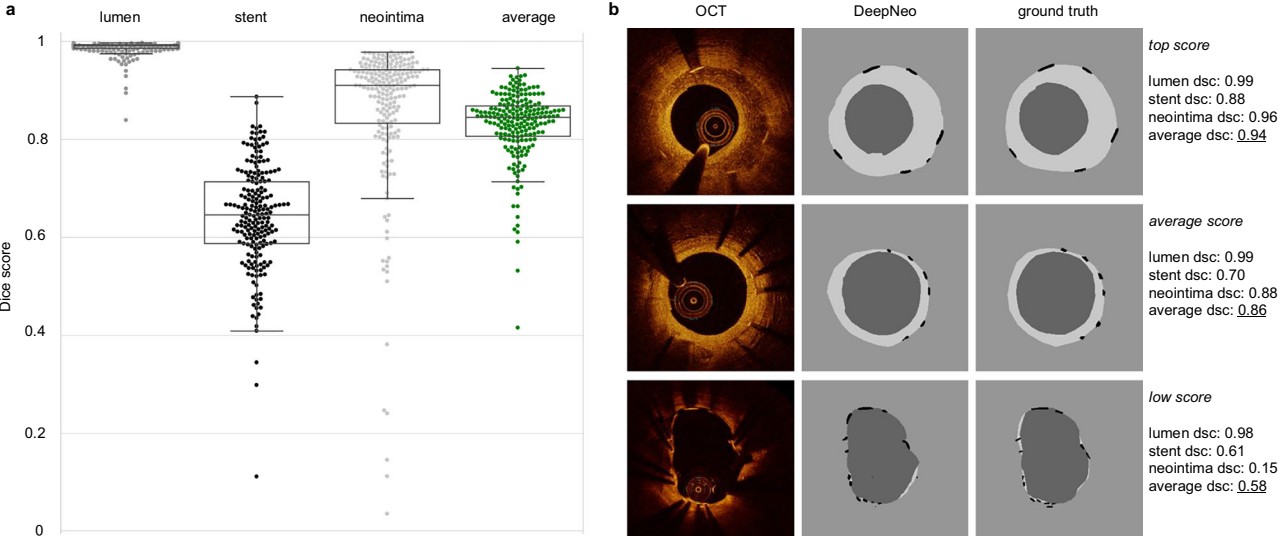

**Fig. 2 | DeepNeo accurately segments lumen, stent, and neointima.** Beeswarm plot with boxplot, with median values (central horizontal black line), boxes extend from the 25th to the 75th percentile of scores generated by 5-fold cross-validation on *n* = 305 images from 42 patients. Whiskers represent the range from the minimum to the maximum values within 1.5 times the interquartile range from the 25th and 75th percentiles (**a**). Good, average, and low-performing samples with respect to the average Dice score (dsc) of an image (**b**). The Dice score is calculated as the area of overlap between labeled ground truth and prediction, ranging from 0 to 1 (0 indicating no overlap and 1 complete overlap between prediction and ground truth).

Confusion matrices shown in Fig. 4 demonstrate the performance of DeepNeo similar to that of clinical experts. Notably, DeepNeo rarely misses diseased frames, indicating its reliability in detecting heterogenous neointima and neoatherosclerosis. Disagreement between DeepNeo and expert A, as well as inter-expert disagreement, is highest for these challenging neointimal types, with expert B and expert C sometimes leaning towards homogenous labeling. Additional examples of DeepNeo's automated analysis are shown in Fig. 5. Analysis of failed predictions reveals that

shadowing and missing stent struts were the two major sources of misclassification, as shown in Supplemental Fig. S2.

### Animal model for neointima classification

Co-registered histopathology demonstrates a high degree of concordance between DeepNeo's predictions and the underlying tissue characteristics, as illustrated in Fig. 6. DeepNeo achieves an accuracy of 0.87 and a macro F1-Score of 0.78. Specifically, DeepNeo accurately identifies neointimal foam

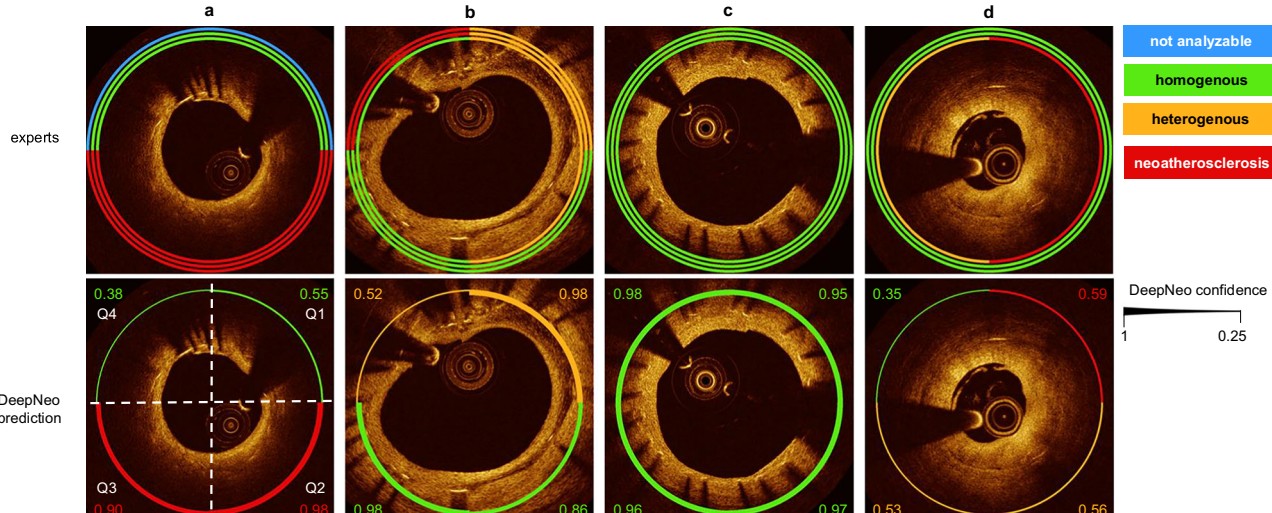

**Fig. 3 | Comparison of manual expert classification and prediction by DeepNeo.** Manual annotation of neointimal tissue type by three different observers is visualized by three separate circular lines for four sample cases (**a–d**). Please note that high interobserver agreement corresponds to a high prediction confidence (quadrants 2 and 3 in **a**, quadrant 3 in **b**, all quadrants in **c** with respective thick prediction lines). In contrast, interobserver disagreement corresponds to lower confidence regarding tissue prediction, visualized by a thinner prediction line. Confidence is computed using test-time augmentations, temperature sharpening the class probabilities, and normalizing to get a calibrated probability distribution.

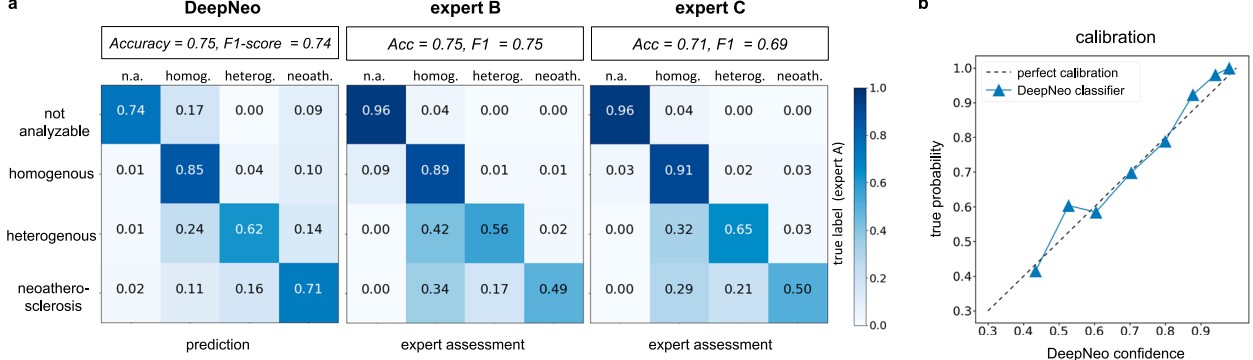

**Fig. 4 | Performance of DeepNeo and human experts.** Confusion matrices for the performance of DeepNeo and experts B and C with labels by expert A taken as ground truth (**a**). Note that automated analysis by DeepNeo is similar to the inter-expert variability. $N = 420$ (not analyzable: 23, homogenous: 186, heterogenous: 117, neoatherosclerosis: 94). Calibration of DeepNeo (**b**): the probability of predicted class (x-axis) vs. true probability (y-axis). To calculate true probability, the samples are split into 10 equally sized bins according to the predicted probability of a sample. The true probability for a bin is then calculated by dividing the number of true predictions by the number of samples.

cells and fibrin deposition as neoatherosclerosis or heterogeneous, while categorizing healthy neointima with an abundance of smooth muscle cells as homogeneous. It is worth noting that DeepNeo achieves these results despite never being trained on rabbit images, demonstrating its robustness and applicability across species and acquisition setups.

## Clinical cases

Figure 7 displays how DeepNeo is applied in two clinical cases of patients who underwent clinically indicated OCT imaging after PCI at the German Heart Center. Neointimal thickness and lumen radius are quantified in a standardized manner by DeepNeo, along with the neointimal tissue composition at pullback level. The visualizations provided by DeepNeo can guide clinicians to identify critical parts of the OCT pullback, which in turn enables a reliable and prompt first impression of the patients. The application of DeepNeo in these clinical cases highlights its potential in improving the standardization and efficiency of intravascular OCT imaging.

## DeepNeo as an open-access tool

We release DeepNeo as an open-access tool, providing a valuable resource for researchers to rapidly analyze intravascular OCT images of stented patients. By making this freeware tool accessible to all, regardless of geographic location or financial resources, we hope to promote collaboration and accelerate progress toward better patient outcomes. The tool is built using Gradio, a freeware software, that can be run locally. The trained models will be made available to all researchers after signing a usage agreement, usage for diagnosis will be excluded. As demonstrated in Fig. 8, DeepNeo offers a user-friendly interface that allows for easy access and analysis of intravascular OCT images. Users can simply upload their anonymized OCT pullback as a DICOM image or zip file, with the tool providing subsequent analysis. A detailed quadrant-level analysis as well as aggregated statistics over the whole pullback can be downloaded. Furthermore, the tool has the capability to determine the starting and ending points of the stent through predicted segmentation masks and subsequent post-processing techniques.

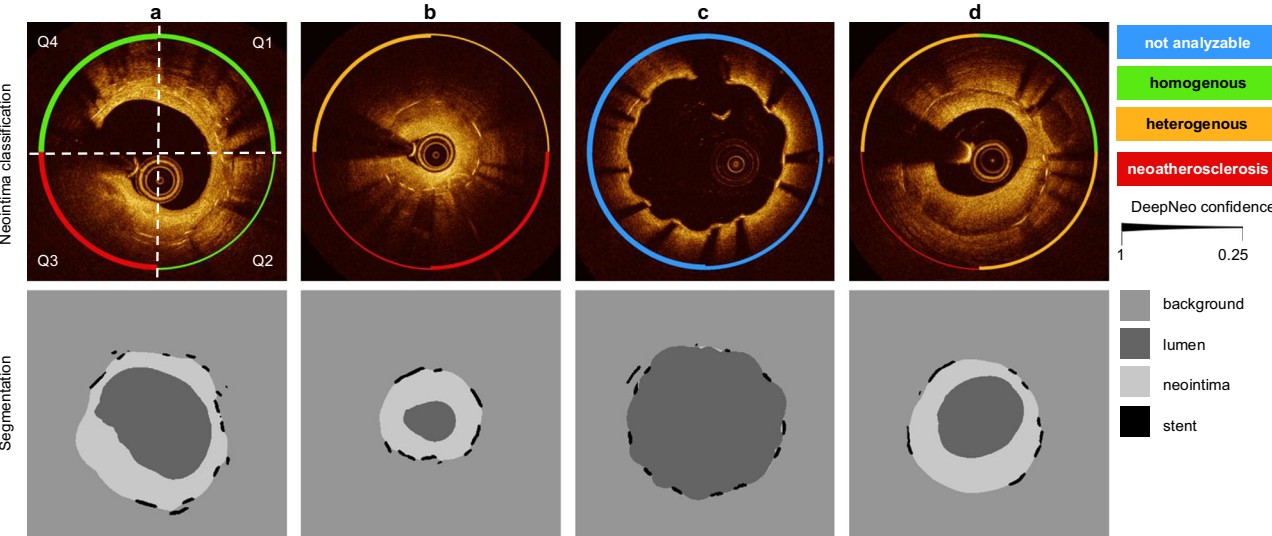

**Fig. 5 | Examples of automated analysis by DeepNeo.** Accurate segmentation and prediction of neointimal tissue characteristics on quadrant level. **a** Predominant homogenous neointima with foam cells in Q3. **b** Heterogenous neointima in Q1 and Q4 with foam cell infiltration in Q2 and Q3. **c** No neointima present. **d** Mixture of homogeneous and heterogenous neointima as well as possible neoatherosclerosis. Note the low confidence in **b** (Q1 and Q3) and **d** (Q3), reflecting the difficulty in differentiating heterogeneous neointima from neoatherosclerosis in some cases. Lower row: automated segmentation of lumen, neointima and stent struts.

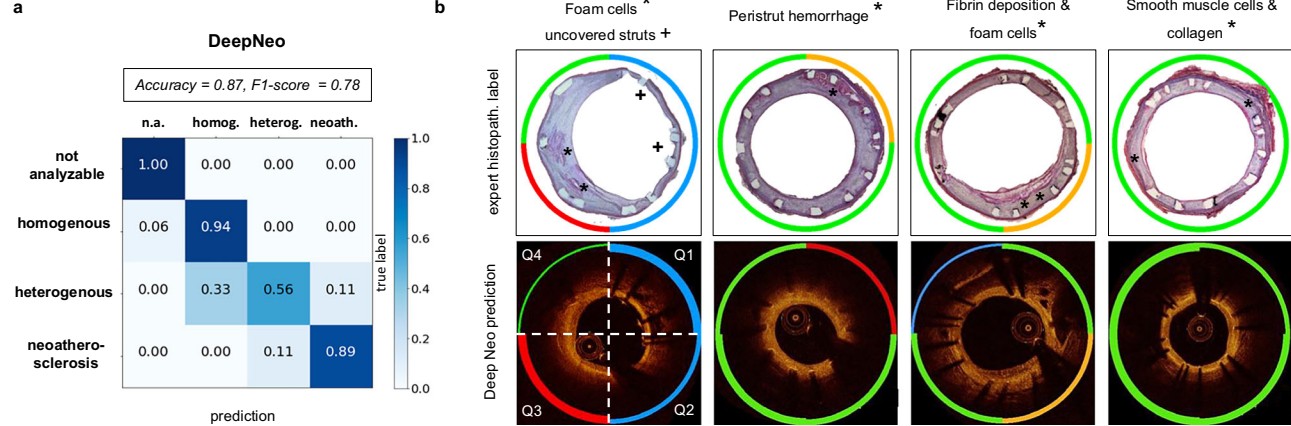

**Fig. 6 | Correlation of tissue prediction by DeepNeo with histopathological findings in rabbits.** **a** Confusion matrix of DeepNeo with histopathological based labels of *n* = 60 quadrants stemming from 12 rabbits. **b** Representative examples from a rabbit model of neoatherosclerosis with hematoxylin–eosin staining, revealing underlying neointimal tissue characteristics with areas of interest marked with *, respectively +. DeepNeo-based analysis of co-registered OCT frames showed overall good agreement between histopathological findings and AI-based tissue prediction, classifying the tissue into not analyzable (blue), homogenous (green), heterogeneous (orange), and neoatherosclerosis (red). Line thickness represents DeepNeo certainty, where high certainty corresponds to a thicker line.

## Discussion

To the best of our knowledge, DeepNeo is the first fully automated deep learning-based algorithm for characterization of vascular healing after PCI using OCT imaging. The algorithm features segmentation of the vessel lumen, neointimal area, and stent struts, enabling further automated morphometric analysis as well as rapid detection and quantification of uncovered stent struts. It also classifies neointimal tissue into healthy (homogenous), diseased (heterogenous or neoatherosclerosis), or not analyzable with high accuracy, matching the performance of human observers. Moreover, its accuracy was confirmed through the analysis of OCT-pullbacks co-registered with histopathology from an animal model of neoatherosclerosis.

With millions of PCIs performed globally every year[26], there is a pressing need for effective diagnostic and therapeutic strategies to ensure optimal patient outcomes in the long term. Intravascular imaging with optical coherence tomography enables high-resolution imaging of stented lesions with detailed visualization of the neointima. Several studies have demonstrated that subjects with neointima characterized as heterogenous have a higher risk of clinical events compared to subjects with homogenous neointima[13,27]. Additionally, heterogenous neointima might also reflect a more atherogenic milieu per se, as it is associated with the progression of native atherosclerosis as well[28]. Hence, heterogenous neointima following stent implantation could be regarded as a surrogate marker for poor arterial healing and adverse clinical outcomes over time. Neoatherosclerosis presents an even more unstable condition[20], being detected in up to one-third of drug-eluting stents[29]. Using OCT, neoatherosclerotic plaque rupture was recently identified as the major underlying cause in patients presenting with very late stent thrombosis[30,31]. Recently, Xhepa et al. demonstrated that a detailed assessment of neointimal tissue characteristics may aid in the selection of dedicated treatment strategies in patients with in-stent restenosis, showing an advantage of DES over DCB in patients with high amounts of non-homogenous frames[32]. Thus, intracoronary imaging with OCT is crucial for following up on patients after PCI with stent implantation, detecting and triaging patients at higher risk of device-related events.

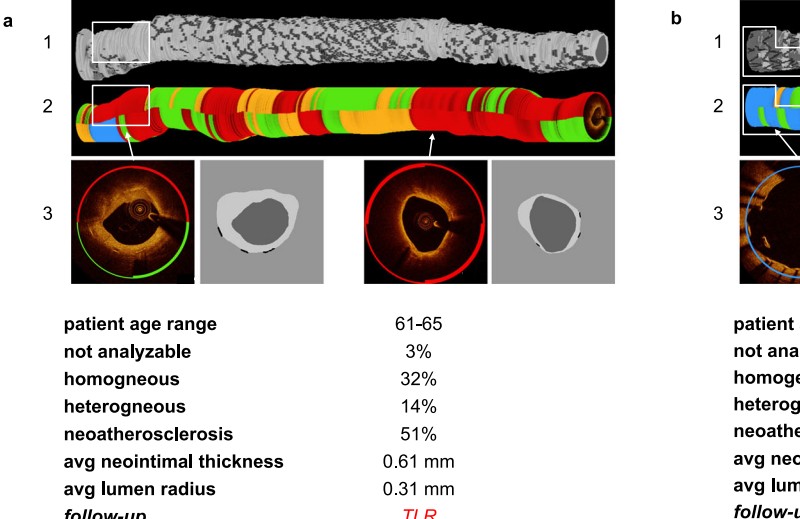
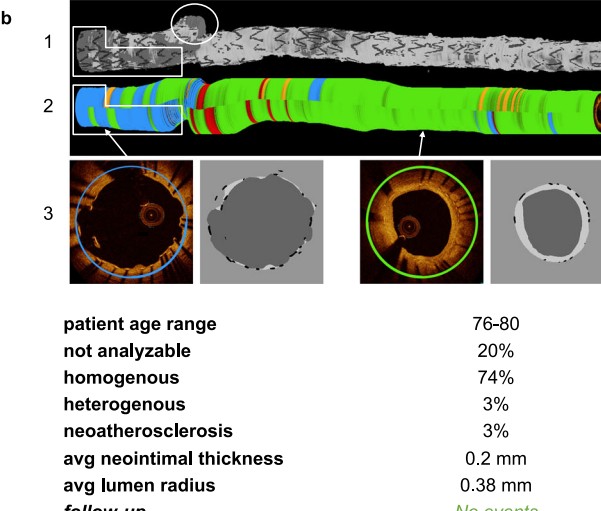

| patient age range | 61-65 |
|---|---|
| not analyzable | 3% |
| homogneous | 32% |
| heterogneous | 14% |
| neoatherosclerosis | 51% |
| avg neointimal thickness | 0.61 mm |
| avg lumen radius | 0.31 mm |
| *follow-up* | *TLR* |

| patient age range | 76-80 |
|---|---|
| not analyzable | 20% |
| homogenous | 74% |
| heterogenous | 3% |
| neoatherosclerosis | 3% |
| avg neointimal thickness | 0.2 mm |
| avg lumen radius | 0.38 mm |
| *follow-up* | *No events* |

**Fig. 7 | Clinical cases.** 3D reconstruction of neointima, lumen and stents (1) as well as 3D reconstruction of neointimal tissue prediction (2) and sample frames (3) from two clinical cases. Quantitative statistics derived from DeepNeo are provided as well. **a** Male with PCI of RCA. OCT 12 months after PCI reveals predominately neoatherosclerotic neointima. During follow-up, the patient underwent target lesion revascularization (TLR) due to in-stent restenosis with unstable angina. **b** Male with PCI of the left anterior descending artery. OCT 12 months after PCI reveals predominantly homogenous neointima. During follow-up, no adverse events occurred. Note how neoatherosclerosis can lead to a loss of signal leading to undetected stent struts (white box in **a**1 and **a**2). Note the correct classification of uncovered stent struts as "not analyzable" (blue line in **b**1 and **b**2) and detection of a side-branch (white circle in **b**1).

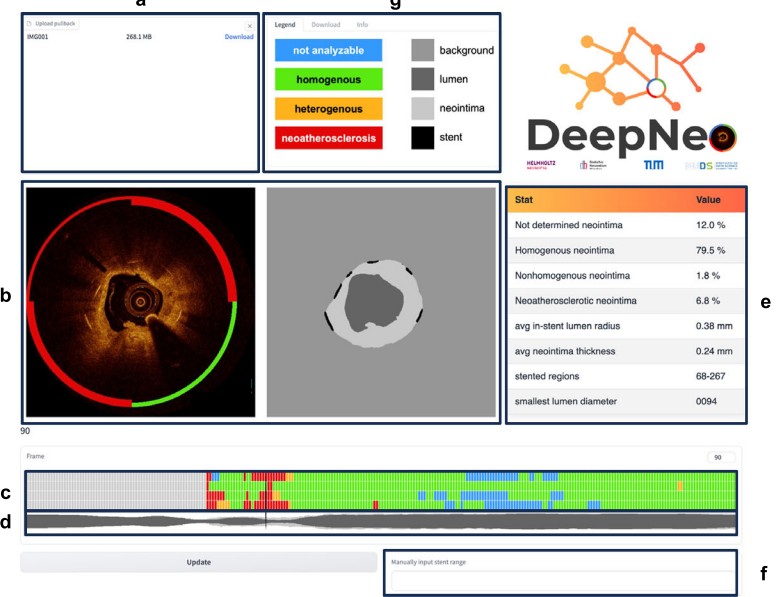
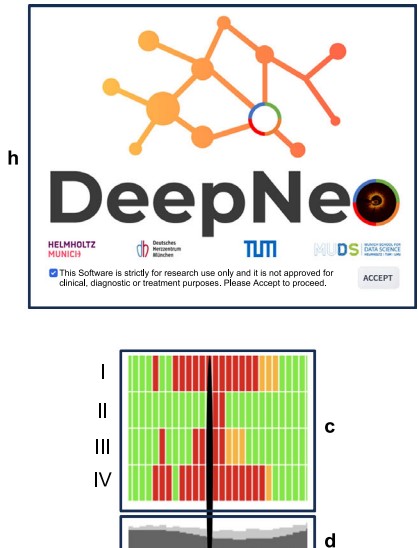

**Fig. 8 | DeepNeo user interface.** The user-friendly interface is designed with several features to facilitate accurate and efficient analysis, including an upload mask (**a**), which allows users to upload OCT pullback images (DICOM or.zip), a visual representation of the current OCT frame with segmentation and neointima prediction (**b**), a schematic view of quadrants (**c**) (top row represents quadrant I, bottom row quadrant IV) and neointima and lumen (**d**) that provides a visual representation of the tissue characteristics, including a slider that enables users to move through the pullback. In addition, the interface includes a pullback analysis (**e**) that provides a detailed analysis of the OCT images and a manual correction feature (**f**) to correct the beginning and end of the stent if necessary. The webtool also allows users to download a detailed analysis of their results and provides an information tab (**g**) for additional guidance. Users are required to accept the research-only use on the welcome page (**h**) before accessing the tool.

However, interpretation of OCT images requires clinical expertise, and analyzing several hundred OCT frames is time-consuming and impractical in busy clinical settings. With an aging population requiring medical attention, the use of deep learning-based algorithms for clinical decision support and hence reduced workload is reasonable and has already been demonstrated in different fields of medicine[33,34]. Previous works have demonstrated the ability to segment and characterize native atherosclerotic lesions using artificial intelligence-enhanced OCT[16,17,35,36]. However, to the best of our knowledge, no study so far has investigated the potential of deep learning to facilitate OCT-based characterization of neointima. We believe that DeepNeo, which allows quick and intuitive, fully automated characterization of the underlying neointima without requiring additional human input, would be useful in following up on vulnerable patients. DeepNeo, in combination with DeepAD[17], our previously published work

on the detection of native atherosclerotic lesions, provides interventional cardiologists with a useful toolbox for facilitating OCT interpretation on native as well as stented segments.

As a limitation of our study, we did not differentiate between layered neointima and heterogenous neointima, as such a distinction would have reduced the sample size for each tissue class and adversely affected the performance of DeepNeo. While cross-validation would have been advantageous for classification as well, we are pleased to have 416 labels annotated by three independent experts, which we believe are sufficient for a robust evaluation. The high performance of the model on animal frames may be influenced by the limited data available for evaluation. The accuracy of DeepNeo in classifying neoatherosclerosis and heterogenous versus homogeneous neointima (71% and 62% versus 85%) in clinical cases may be partly explained by the increasing complexity of neointimal tissue. Homogenous neointima typically displays a simple and uniform appearance, whereas neoatherosclerosis, characterized by foam cells, calcification, or fibroatheroma, exhibits a more diverse and complex aspect that poses a challenge for accurate classification. It is worth emphasizing that a comparable reduction in performance is observed in human experts, indicating that the task of distinguishing between different types of neointimal tissue is inherently challenging. This observation suggests that the reduction in the accuracy of DeepNeo is mainly not due to a failure of the algorithm but rather a reflection of the complexity of the task. Additionally, splitting a frame into four quadrants might create ambiguous cases, such as when portions of a quadrant are more severely diseased, making classification challenging. However, it is noteworthy that misclassifications of neoatherosclerosis as heterogenous neointima or vice versa may still be considered acceptable, as both conditions are indicative of diseased tissue that requires further attention. Moreover, the identification of any diseased tissue through automated analysis can help alert clinicians to potential issues, prompting further investigation and intervention where necessary. In rare circumstances, such as inadequate contrast medium or highly atypical cases, the model may encounter difficulties. However, due to the calibrated model, those cases should result in low-confidence predictions and could be flagged for further inspection. Thus, even with some degree of misclassification, DeepNeo is a valuable tool in the detection and characterization of neointimal tissue in patients after PCI.

DeepNeo offers a fast, reliable, and standardized approach to neointima characterization after PCI. Its performance closely matches that of clinical experts, and its well-calibrated predictions ensure interpretability, making acceptance by clinicians more likely. By reducing the need for time-intensive manual interpretation, DeepNeo has the potential to streamline clinical workflows and support interventional cardiologists in making more informed decisions. Its open-access nature fosters collaboration and accelerates research in vascular healing, while its applicability across species and imaging setups underscores its robustness. As deep learning continues to integrate into medical practice, tools like DeepNeo will be instrumental in enhancing diagnostic precision, improving patient outcomes, and paving the way for further AI-driven innovations in cardiovascular medicine.

## Data availability
The data underlying this article will be shared on reasonable request to the corresponding authors. Source data used to create Fig. 2 can be found in supplemental data 1. Source data used to create Fig. 4 and supplemental Fig. 1 can be found in supplemental data 2. Source data used to create Fig. 6 can be found in supplemental data 3. Models can be downloaded after approval on Zenodo[37].

## Code availability
The full code can be found on github.com/ValentinKoch/DeepNeo[38].

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

## Acknowledgements

This work was supported through a scientific grant by the German Cardiac Society (Grant Number: 16/2020) and by the Helmholtz Association under the joint research school "Munich School for Data Science—MUDS". It was also supported by the BMBF-funded de.NBI Cloud within the German Network for Bioinformatics Infrastructure (de.NBI) (031A532B, 031A533A, 031A533B, 031A534A, 031A535A, 031A537A, 031A537B, 031A537C, 031A537D, 031A538A). C.M. has received funding from the European Research Council under the European Union's Horizon 2020 research and innovation program (grant agreement number 866411). T.K. received funding from the German Research Foundation (DFG) as part of a research project (KE 2116/4-1) and the Heisenberg program (KE 2116/5-1). Additional funding was received from the Translation and Innovation grant from Helmholtz Munich.

## Author contributions

V.K. designed and implemented the program code, designed and evaluated the experiments, created the visualizations for the figures, and was a primary contributor to the writing of the paper. O.H. processed data and initiated the collaboration. E.S. helped revise the manuscript and run additional experiments. A.A., M.S., E.X., J.W., S.C., S.K., T.K., H.S., F.V., T.R., T.L., A.K., H.S., P.N. collected and curated the underlying data. J.A.S., M.J. and C.M. supervised the project. A.A. and M.S. labeled parts of the classification data. E.B. labeled the majority of the segmentation data. P.N. was the primary annotator, initiated the collaboration, designed and evaluated the experiments, designed the figures, and was a primary contributor to the writing of the paper. All authors played a crucial role in the review and refinement of the manuscript, ensuring its accuracy and coherence.

## Funding

## Competing interests

V.K. and P.N. are named inventors on a patent application describing the technology of this manuscript. T.K. is named inventor on a patent application for the prevention of restenosis after angioplasty and stent implantation outside the submitted work. T.K. received lecture fees from Bayer, Abbott, Bristol-Myers Squibb, and Astra-Zeneca which are unrelated to this work. All other authors declare that they have no competing interests.
