## [Transparent Peer Review file · Communications Medicine]

Deep learning model DeepNeo predicts neointimal tissue characterization using optical coherence tomography

Corresponding Author: Dr Carsten Marr

Version 0:

Reviewer comments:

Reviewer #1

(Remarks to the Author)

The manuscript described a method that utilizes deep learning models to address the task of classifying neointimal tissue. This work tested four types of prediction results, among which the outcomes demonstrated considerable performance. Additionally, 'DeepNeo' has the potential to be a valuable tool for post-procedure observation and diagnosis following percutaneous coronary intervention (PCI).

However, attention should be paid to the following comments:

1. The manuscript's logical structure needs revision for clearer presentation.
2. There are grammatical errors in some sentences.
3. The paper briefly introduces and applies two deep learning models for the research. However, there is a lack of comparison with previous studies and a detailed rationale for choosing UNet++ and ResNet as the final approaches. Additionally, the overall process is not adequately described.
4. The 'Conclusion' section should discuss the approach to handling complex situations and potential limitations of this research.
5. The four-quadrant method may overlook some local details. The manuscript should also discuss how to address this issue if it arises.

The manuscript described a method that utilizes deep learning models to address the task of classifying neointimal tissue. This work tested four types of prediction results, among which the outcomes demonstrated considerable performance. Additionally, 'DeepNeo' has the potential to be a valuable tool for post-procedure observation and diagnosis following percutaneous coronary intervention (PCI).

However, attention should be paid to the following comments:

1. The manuscript's logical structure needs revision for clearer presentation.
2. There are grammatical errors in some sentences.
3. The paper briefly introduces and applies two deep learning models for the research. However, there is a lack of comparison with previous studies and a detailed rationale for choosing UNet++ and ResNet as the final approaches. Additionally, the overall process is not adequately described.
4. The 'Conclusion' section should discuss the approach to handling complex situations and potential limitations of this research.
5. The four-quadrant method may overlook some local details. The manuscript should also discuss how to address this issue if it arises.

Reviewer #2

(Remarks to the Author)

This study aimed to develop a deep-learning method to analyze OCT images and classify them as homogeneous, heterogenous, and neo-atherosclerosis using deep learning models. Although the number of patients was limited, the novelty is high, and the method is straightforward. In addition, the experiments were set up properly. The authors claimed

that the developed method could be utilized in many clinical scenarios. However, there are many critical points that need to be addressed to improve the quality of the manuscript. Please see below.

Keywords

- The terms 'vascular healing' and 'computer vision' are inappropriate. Please include neointima and DeepNeo instead.

Methods

- Line 103: This section should only include information related to neointima classification. Please create a sub-section (Data acquisition) and include relevant information (e.g., number of patients, acquisition protocol, etc.). Also, please move lines 98-101 to this section.
- Line 104: Please add the number of frames (i.e., ... 1,148 OCT images from 92 patients ...)
- Line 109: It is better to say 'every 1 mm.'
- Line 110: Please remove '... A total of 1148 frames from 92 pullbacks were analyzed.'
- Line 115, ... one of four ...: There are three only. Please add 'not analyzable.'
- Line 122-126: So, the number of frames for training, validation, and internal testing are 936, 108, and 104, respectively. Please rephrase it with the correct numbers of training, validation, and testing sets. Also, please ensure that there was no data overlap between training/validation/testing groups. Most importantly, I strongly recommend 5-fold cross validation for neointima classification. It is plausible because the number of cases for training is small.
- Line 150: The segmentation of neointima, lumen, and stent struts should be presented before neointimal classification since this step is a prerequisite step for classification. Therefore, the order of method section should be data acquisition, segmentation, neointima classification, animal model for neointima classification, and algorithm architecture.
- Line 151: The author only used 90 images for model development. This is not enough, and therefore, the results are not convincing. Please increase the number of cases to at least 300.
- Line 170: The term '(i) segment features of interest' is incorrect. It should be 'segment lumen, stent struts, and neointima'.
- Line 176: Please clarify why model calibration is required and important. If possible, please include results with/without this step. This should be a simple but powerful way to show the rationale.

Results

- Line 195, ... indicating that DeepNeo is well-calibrated.: This statement is not convincing. It is hard to understand why this step was important. Please clarify the rationale.
- Line 211: The accuracy difference between the testing and animal testing set is 12%. Normally, if there is a big performance difference between internal and external testing sets, the network tends to be over-trained. Please clarify the appropriate reason in the Discussion. One reason might be the small number of animal cases (15 frames).
- Line 218: This should be reported before neointima classification.

Discussion

- Line 266: This should be the first main finding.
- The figure/table captions are not available.

Version 1:

Reviewer comments:

Reviewer #2

(Remarks to the Author)

I appreciate the authors' revisions. All my concerns have been addressed.

Reviewer #3

(Remarks to the Author)

The authors have addressed the comments of reviewer #1 in a satisfactory way.

Point by Point reply

Reviewer comments are written in **bold**, our reply in *italic*, changes to the manuscript are in quotation marks.

We mark sentences with changes in **green colour in the manuscript** as well as in the supplemental material. Sentences with only minor grammar changes were not marked.

Reviewer #1 (Remarks to the Author):

The manuscript described a method that utilizes deep learning models to address the task of classifying neointimal tissue. This work tested four types of prediction results, among which the outcomes demonstrated considerable performance. Additionally, 'DeepNeo' has the potential to be a valuable tool for post-procedure observation and diagnosis following percutaneous coronary intervention (PCI).

However, attention should be paid to the following comments:

1. The manuscript's logical structure needs revision for clearer presentation.

We thank the reviewer for this remark and revised the structure, so that the presentation is now more straightforward.

2. There are grammatical errors in some sentences.

We fixed the grammatical errors.

3. The paper briefly introduces and applies two deep learning models for the research. However, there is a lack of comparison with previous studies and a detailed rationale for choosing UNet++ and ResNet as the final approaches. Additionally, the overall process is not adequately described.

We thank the reviewer for this important feedback.

a) We added more references to previous related studies to the revised discussion^{1,2}, where we now write:

“Previous works have demonstrated the ability to segment and characterize native atherosclerotic lesions using artificial intelligence-enhanced OCT¹⁻⁴. However, to the best of our knowledge, no study so far has investigated the potential of deep learning to facilitate OCT-based characterization of neointima.”

b) We also added experiments motivating the choice of Unet++ as the segmentation network and compared it to the standard Unet and DeepLabv3 approaches. Its performance is slightly better than Unet and better than DeepLabv3, and it converged faster. We added the according table in the supplementary material, where we now write:

“As the segmentation networks, we evaluate a standard Unet, Unet++ and DeepLabv3. We observe that Unet++ performs slightly better on average than Unet, which both perform better than DeepLabv3 as can be seen in supplementary table 1. As the best model, the Unet++ with Resnet18 backbone is used as the segmentation network”

	lumen	stent	neointima	average
Unet	0.986±0.025	0.658±0.101	0.863±0.135	0.837
Unet++	0.986±0.021	0.660±0.100	0.863±0.138	0.838
DeepLabv3	0.986±0.014	0.603±0.110	0.868±0.138	0.820

c) We added new experiments to the supplemental material, where we show that ResNet performs better than the two other state-of-the art models (vision transformer and swin transformer). We took the smallest pytorch implementation of each model, both having at least as many parameters as Resnet18. We now write in the supplementary materials and methods in the Details of classification network and calibration section:

“We found that for the classification task small models suffice. Among those, a torchvision ResNet18 showed the best performance compared to other state-of-the art networks with the same or more parameters, implemented in the torchvision library. Namely, we compare the ResNet18 performance to ViT-B (smallest torchvision vision transformer) and Swin_T (smallest torchvision Swin transformer) (see supplementary table 2). Note that for this comparison we measure the basic performance without test time augmentation, temperature sharpening, and without considering neighboring frames. As Resnet18 showed the best performance, it was selected as the network for DeepNeo.”

	accuracy	f1-score
Resnet18	0.690	0.697
ViT-B	0.557	0.574
Swin-S	0.443	0.272

4. The 'Conclusion' section should discuss the approach to handling complex situations and potential limitations of this research.

We appreciate the reviewer's remark and have included an extensive discussion on the limitations of our approach in complex situations in the Conclusions section of the revised manuscript:

“As a limitation of our study, we did not differentiate between layered neointima and heterogenous neointima, as such distinction would have reduced the sample size for each tissue class and adversely affected the performance of DeepNeo. Furthermore, the high performance of the model on animal frames may be influenced by the limited data available for evaluation.”

“Additionally, splitting a frame into four quadrants might create ambiguous cases, such as when portions of a quadrant are more severely diseased, making classification challenging.”

“In rare circumstances, such as inadequate contrast medium or highly atypical cases, the model may encounter difficulties; however, due to the calibrated model, those cases should result in low confidence predictions and could be flagged for further inspection.”

5. The four-quadrant method may overlook some local details. The manuscript should also discuss how to address this issue if it arises.

We acknowledge that edge cases remain, although the four-quadrant method has been described and published before⁵⁻⁷. We added a clarification that in edge cases the annotators were instructed to label the more severe class in the Method section:

“In quadrants with more than one tissue type, the most severe neointimal tissue type was scored”.

In the Conclusion section, we added:

“Additionally, splitting a frame into four quadrants might create ambiguous cases, such as when portions of a quadrant are more severely diseased, making classification challenging.”

Reviewer #2 (Remarks to the Author):

This study aimed to develop a deep-learning method to analyze OCT images and classify them as homogeneous, heterogenous, and neo-atherosclerosis using deep learning models. Although the number of patients was limited, the novelty is high, and the method is straightforward. In addition, the experiments were set up properly. The authors claimed that the developed method could be utilized in many clinical scenarios. However, there are many critical points that need to be addressed to improve the quality of the manuscript. Please see below.

Keywords

- The terms ‘vascular healing’ and ‘computer vision’ are inappropriate. Please include neointima and DeepNeo instead.

We thank the reviewer for this remark and changed the keywords accordingly.

Methods

- **Line 103:** This section should only include information related to neointima classification. Please create a sub-section (Data acquisition) and include relevant information (e.g., number of patients, acquisition protocol, etc.). Also, please move lines 98-101 to this section.

We thank the reviewer for this excellent suggestion and applied the requested changes. The new subsection now reads:

Data Acquisition

1148 OCT images from 92 patients who underwent clinically-indicated coronary angiography and in-stent intravascular imaging with OCT at the German Heart Center Munich were collected. OCT imaging was performed according to current guidelines¹⁷ using a commercially available OCT system (Abbott Vascular, Santa Clara, CA). The baseline characteristics of patients are provided in Table 1. “

We think however that lines 98-101 should stay outside this section, as the information about filing of a patent does not fit into the Data acquisition subsection.

- **Line 104:** Please add the number of frames (i.e., ... 1,148 OCT images from 92 patients ...)

We applied the requested changes and now write in the Data Acquisition section:

“1148 OCT images from 92 patients who underwent clinically-indicated coronary angiography and in-stent intravascular imaging with OCT at the German Heart Center Munich were collected.”

- **Line 109:** It is better to say ‘every 1 mm.’

We changed the wording and included the distance in mm:

“Manual quadrant annotation of OCT frames every 1 mm (every fifth frame) was performed for all pullbacks or in adjacent suitable frames, when image quality was insufficient.”

- **Line 110: Please remove ‘... A total of 1148 frames from 92 pullbacks were analyzed.’**

We removed this passage.

- **Line 115, ... one of four ...: There are three only. Please add ‘not analyzable.’**

Thanks for pointing this out. We corrected the sentence and added the “not analyzable” class.

- **Line 122-126: So, the number of frames for training, validation, and internal testing are 936, 108, and 104, respectively. Please rephrase it with the correct numbers of training, validation, and testing sets. Also, please ensure that there was no data overlap between training/validation/testing groups. Most importantly, I strongly recommend 5-fold cross validation for neointima classification. It is plausible because the number of cases for training is small.**

Thank you for raising this important point. We double checked and can confirm that there is indeed no overlap between the different splits. We additionally rephrased the number of frames accordingly.

Unfortunately only parts of the dataset are labeled by three independent experts, so in order to make a comparison between DeepNeo and human observers we were not able to do 5-fold cross validation. However, we argue that 416 test quadrants are a reasonable size and sufficient to draw conclusions about the performance of DeepNeo.

We discuss this limitation in the revised version of the manuscript, where we now write in the discussion:

“While cross-validation would have been advantageous for classification as well, we are pleased to have 416 labels annotated by three independent experts, which we believe are sufficient for a robust evaluation.”

- **Line 150: The segmentation of neointima, lumen, and stent struts should be presented before neointimal classification since this step is a prerequisite step for classification. Therefore, the order of method section should be data acquisition, segmentation, neointima classification, animal model for neointima classification, and algorithm architecture.**

Thank you for the suggestions which help clarify the process. The manuscript was adjusted accordingly.

- **Line 151: The author only used 90 images for model development. This is not enough, and therefore, the results are not convincing. Please increase the number of cases to at least 300.**

Following the reviewer's suggestion, we increased the dataset size to 305 images with the help of a new annotator. Notably, the performance increased slightly from 0.98 to 0.99 dice score for lumen and from 0.84 to 0.86 for neointima. The slight decrease of performance of the stent class (0.71 to 0.66) might be contributed to a slightly different annotation style of the new annotator, who had a tendency to annotate stent struts more thinly.

We updated the relevant text passages and the figures accordingly.

- **Line 170: The term '(i) segment features of interest' is incorrect. It should be 'segment lumen, stent struts, and neointima'.**

We changed the wording accordingly.

- **Line 176: Please clarify why model calibration is required and important. If possible, please include results with/without this step. This should be a simple but powerful way to show the rationale.**

We thank the reviewer for this suggestion and agree that this is indeed helpful. We added a supplementary figure 2, showing that without calibration, the model's confidence does not correlate well with the true probability. It tends to be overly confident, which is a well known problem in deep learning. We add the motivation and the reference to the figure in the results section and write

"In supplemental figure 2 we show the need for calibration: the uncalibrated version of our model tends to be overly confident, and the correlation between the (uncalibrated) confidence and the true probability is poor."

Results

- **Line 195, ... indicating that DeepNeo is well-calibrated.: This statement is not convincing. It is hard to understand why this step was important. Please clarify the rationale.**

Thank you for the comment. We added supplementary figure 2, showing that calibration is needed: It shows that when no calibration is applied, the true probability does not correlate well with the confidence of the model. Well-calibrated models provide reliable uncertainty estimates, allowing experts to identify and review less confident cases more closely.

Supplemental Figure 2: Model calibration improves correlation between confidence and true probability. Uncalibrated model (left) vs calibrated model used in DeepNeo (right).

- **Line 211: The accuracy difference between the testing and animal testing set is 12%. Normally, if there is a big performance difference between internal and external testing sets, the network tends to be over-trained. Please clarify the appropriate reason in the Discussion. One reason might be the small number of animal cases (15 frames).**

Indeed, a typical sign of overtraining would be a higher in-domain performance, which is not the case here. As the higher performance was achieved on the external test set, we agree that the reason is likely to be the small number of cases. We added this to the discussion, where we now write:

“The high performance of the model on animal frames may be influenced by the limited data available for evaluation.”

- **Line 218: This should be reported before neointima classification.**

Thanks for the suggestion, we changed the script accordingly.

Discussion

- **Line 266: This should be the first main finding.**

We agree and changed the manuscript accordingly.

- **The figure/table captions are not available.**

We are sorry to hear that. We made sure to fill out the description of the figures in the upload mask, should that not work we additionally added a document with the corresponding figure and table captions.

References

1. Chu, M. *et al.* Artificial intelligence and optical coherence tomography for the automatic characterisation of human atherosclerotic plaques. *EuroIntervention* **17**, 41–50 (2021).
2. Lee, J. *et al.* Segmentation of Coronary Calcified Plaque in Intravascular OCT Images Using a Two-Step Deep Learning Approach. *IEEE Access* **8**, 225581–225593 (2020).
3. Fedewa, R. *et al.* Artificial Intelligence in Intracoronary Imaging. *Curr. Cardiol. Rep.* **22**, 46 (2020).
4. Holmberg, O. *et al.* Histopathology-Based Deep-Learning Predicts Atherosclerotic Lesions in Intravascular Imaging. *Front Cardiovasc Med* **8**, 779807 (2021).
5. Xhepa, E. *et al.* Qualitative and quantitative neointimal characterization by optical coherence tomography in patients presenting with in-stent restenosis. *Clin. Res. Cardiol.* **108**, 1059–1068 (2019).
6. Nicol, P. *et al.* Validation and application of OCT tissue attenuation index for the detection of neointimal foam cells. *Int. J. Cardiovasc. Imaging* **37**, 25–35 (2021).
7. Xhepa, E. *et al.* Clinical outcomes by optical characteristics of neointima and treatment modality in patients with coronary in-stent restenosis. *EuroIntervention* **17**, e388–e395 (2021).